# Exploring Efficient Foundational Multi-modal Models for Video Summarization

## Abstract

Foundational models are able to generate text outputs given prompt instructions and text, audio, or image inputs. Recently these models have been combined to perform tasks on video, such as video summarization. Such video foundation models perform pre-training by aligning outputs from each modality-specific model into the same embedding space. Then the embeddings from each model are used within a language model, which is fine-tuned on a desired instruction set. Aligning each modality during pre-training is computationally expensive and prevents rapid testing of different base modality models. During fine-tuning, evaluation is carried out within in-domain videos where it is hard to understand the generalizability and data efficiency of these methods. To alleviate these issues we propose a plug-and-play video language model. It directly uses the texts generated from each input modality into the language model, avoiding pre-training alignment overhead. Instead of fine-tuning we leverage few-shot instruction adaptation strategies. We compare the performance versus the computational costs for our plug-and-play style method and baseline tuning methods. Finally, we explore the generalizability of each method during domain shift and present insights on what data is useful when training data is limited. Through this analysis, we present practical insights on how to leverage multi-modal foundational models for effective results given realistic compute and data limitations.

## 1 Introduction

Learning to model complex modality data such as documents, images, or videos has been historically done with domain-specific architectures suited to a particular modality and dataset (Vaswani et al., 2017; Liu et al., 2021; Sun et al., 2019). As new problems evolved that require bridging modalities, such as image or video question answering, multi-modal architectures have been used to learn joint representations between modalities for a specific end task (Tan & Bansal, 2019; Alayrac et al., 2022). Most recently large foundational models have emerged to improve domain-specific tasks with large-scale pre-training. Foundation models from multiple modalities have been combined and fine-tuned toward specific multi-modal tasks. Due to the large scale of these models and datasets, there remain fundamental questions about their computational requirements and generalizability once they are fine-tuned.

**Computation requirements** Compute for multi-modal video language models are typically conducted in multiple stages (Lyu et al., 2023; Zhao et al., 2023). In the *first stage*, the embeddings produced from models of different modalities are aligned to generate a unified representation for the downstream large language models (LLMs). In videos, this involves aligning, image, audio, or text modalities to one another. However, with the frequency of new foundational model releases, it becomes impractical to realign all modalities when an improved foundational model is introduced.

In addition to modality alignment, there is the *second stage* fine-tuning of the LLM or individual modality models on the domain-specific task carried out. Such tuning is the de facto method to achieve state-of-the-art (SOTA) performance, while zero-shot methods are tested for generalizability. However these are binary choices, and understanding how training data and compute availability scale with performance is yet to be deeply explored in the multi-modal domain.

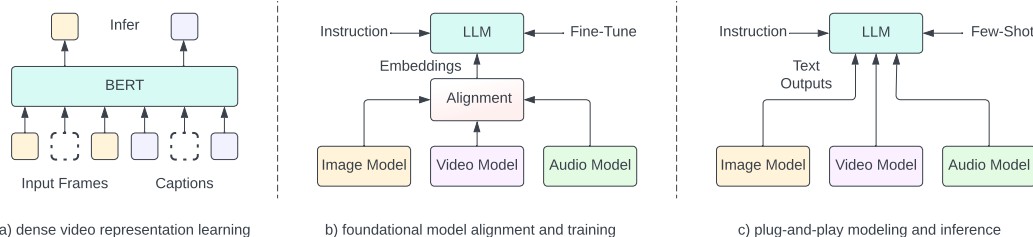

Figure 1: In a) video temporal representations are learned through inferring frame and caption data in videos. Instead of a single model to infer each modality of a video, in b) each modality is handled by a foundational model. The embeddings are aligned and passed as tokens into an LLM for fine-tuning. We explore the setting in c) where we use the text outputs of each modality to quickly leverage new models and adapt to new domains.

It is important to test an alignment-agnostic, yet computationally efficient solution, where we propose a Plug-and-Play Video Language Model (PP-VLM) to evaluate these settings. This framework directly converts video modalities (metadata, frames, audio, and detected objects) into text representations that can be fed directly into a language model for video-based tasks without any fine-tuning.

**Generalizability** Beyond the tuning strategies of a model within a specific domain, we investigate how such models perform when operating out-of-domain. This is important when adapting large models in low-resource or new application domains where there is minimal training data. In such cases, understanding what model sizes work best, how to leverage existing model prompts, and relevant subsets of training data provide a comprehensive picture of domain generalizability.

Given current multi-modal LLM approaches, we explore modifications to support the interchangeability of individual model components while providing guidance on how to efficiently adapt to a target task. We investigate these contributions by:

- Illustrating a simple framework to compose foundational models from different modalities within an open LLM to effectively integrate new models.
- Understanding what input modalities and model sizes are appropriate given a specific domain, and expectations of how multi-modal models adapted to one domain will perform in another.
- Investigating how to fine-tune multi-modal models and how they compare to few-shot approaches.

## 2 RELATED WORKS

### 2.1 VIDEO REPRESENTATION LEARNING

For video-level tasks, learning the shared representation of visual, text, and audio features has been key to improving downstream tasks. Recent BERT (Devlin et al., 2018) based masking methods like VideoBERT (Sun et al., 2019) jointly infer masked caption tokens as well as frame image tokens, shown in Figure 1 a). Similarly, BEVT (Wang et al., 2022) jointly predicts image tokens as well as changes within video frames. UniVL (Luo et al., 2020) aligns the video and caption embeddings and attempts to generate the caption, in addition to image and text masking objectives. Video language model (Xu et al., 2021) leverages the base BERT architecture to alternate between random and full masking of captions and frames while providing different self-attention masking strategies for downstream tasks. Merlot (Zellers et al., 2021) aligns captions and frames, learns temporal orderings between frames, and recovers masked captions. This is extended in Merlot Reserve (Zellers et al., 2022) where audio is jointly input with images and captions into a Transformer (Vaswani et al., 2017) where the text and audio signature are inferred. Vid2seq (Yang et al., 2023) operates over longer videos to densely generate a sequence of captions. While these models effectively learn dense video representations, it is hard to efficiently adapt them to instruction following tasks handled by recent generative foundational models.

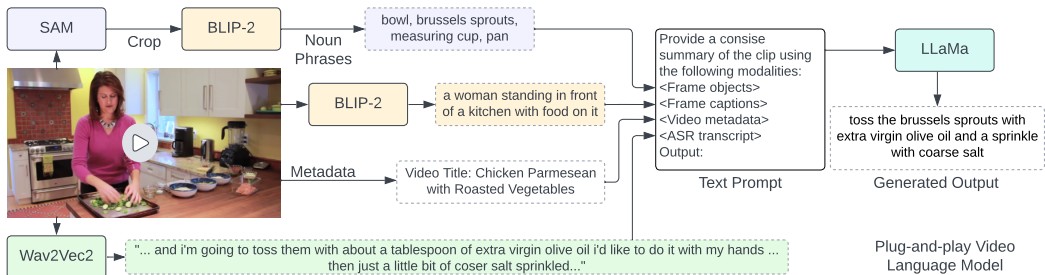

Figure 2: An overview of our model architecture. Each video modality input source generates a text output that is used to generate a prompt fed directly into a language model for instruction following.

## 2.2 FOUNDATIONAL MODEL GENERATION

Large generative text models are being used as the backbone to answer questions about unstructured input texts. These models have been infusing data from other modalities in order to generate responses on multi-modal video data. Macaw-LLM (Lyu et al., 2023) encodes video, audio, and images with corresponding foundational models. The individual modality embeddings are contextualized through cross-attention and are appended with the input prompt text into the final LLM for instruction fine-tuning, illustrated in Figure 1 b). Similarly, ChatBridge (Zhao et al., 2023) also aligns these video modalities but performs an initial pre-training stage to align each modality before performing instruction fine-tuning. These methods require fine-tuning on instruction datasets since they incorporate dense embeddings from each modality. Thus the performance is unclear when the domain shifts to a new distribution of videos and would require expensive re-training if data is available. Instead, we investigate the performance when no alignment is performed and the text output from video modalities is used directly within the LLM for video tasks as shown in Figure 1 c).

## 3 METHOD

To operate over video data we leverage both the visual and the audio modalities that it consists of. These include video frames, object segmentations obtained from frames, audio extracted from the video, and any metadata provided by the video. In contrast to previous video language models, we input each of these intermediate modalities as text representations instead of dense embeddings. The generated texts from the corresponding foundational model are fed into a language model to perform the end task as shown in Figure 2. In such a setting we avoid joint training of the separate modality embeddings along with the language model, allowing us to test different modality models without any re-training. Instead, we leverage the variance of data captured by each foundational model independently on large-scale datasets and apply it directly within our task of interest.

### 3.1 VIDEO MODALITIES TO TEXT

To represent the video as text, we first select the middle frame of the video clip as a visual representation of the clip. We use a BLIP-2 Li et al. (2023b) vision language Transformer to obtain the caption for the frame. While the frame caption obtains a high-level description of the scene, the individual objects in the frame are useful to describe the scene. Using fixed-vocabulary detectors covers common objects, but does not cover the variety of objects that can be observed in different video domains. Instead, we use the Segment Anything Model (SAM) (Kirillov et al., 2023) to obtain segmentation regions within our video frame. Then we crop each segmentation and pass it back to BLIP-2 to caption the patch. Since we are looking for object descriptions, we only keep noun phrases found within each caption as an object pseudo-label. For the clip audio, we use Wav2Vec2 (Baevski et al., 2020), where we directly use the recognized speech as the text output. Finally, any metadata for the video, including its title or categorization classes, is also added to the text prompt.

## 3.2 FEW-SHOT ADAPTATION

We feed the text modalities extracted from the video into LLaMa (Touvron et al., 2023) for instruction following. We are motivated to adapt the video inputs for new tasks without end-to-end fine-tuning over a large number of training samples. Our processed training set $T = \{X, Y\}$ is composed of the text representation outputs from each frozen modality model $X = \{\text{image}_i, \text{metadata}_i, \text{speech}_i, \text{objects}_i\}_{i=1}^N$ along with the desired output text generations $Y = \{\text{outputs}_i\}_{i=1}^N$. In the simplest setting, we select few-shot samples at random $R \subset T$ and construct a prompt that indicates what we want to generate, a description of the modalities, followed by our few-shot samples $R$. Then for each inference example, we append the input text modalities of the video and ask it to generate the corresponding output. Given additional training data, Lu et al. (2021) show that the selection and ordering of few-shot examples makes a large difference in final evaluation performance. Therefore we test alternative strategies to select better prompt examples.

**Greedy Few-shot Search** To select better prompts, we adapt the in-context influences proposed by Nguyen & Wong (2023) where we build our few-shot samples in a greedy fashion from our training set $T$. This is done by selecting a holdout set from the training samples $H \subset T$ for evaluation. Then we select a search subset of training samples that are not within our holdout set $S \subset T \setminus H$. We then find a sample with the search set to add to our running set of previous few-shot examples selected $F = \{X_F, Y_F\}$. In particular, we find the sample that maximizes our generation metric score on the holdout set: $\arg\max_{x_s, y_s \in S} \sum_{x_h, y_h \in H} \text{Score}(y_h, \hat{y_h})$. Here $\hat{y_h}$ is generated from our model given the input few-shot prompt built from $\{X_F \cup \{x_s, x_h\}, Y_F \cup y_s\}$. The optimal sample $x_s^*, y_s^* \in S$ is added to the set of few-shot examples used for the next iteration $F = \{X_F \cup x_s^*, Y_F \cup y_s^*\}$. Note that we fix the holdout set $H$ but re-sample the search set $S$ at each iteration to capture a larger variance of candidate samples. We iterate through this process and greedily build the set of n-shot examples that are used for inference, where $n = |F|$.

## 4 EXPERIMENTAL SETUP

### 4.1 DATASETS

We test our method on video summarization tasks, where given a video clip, the model should provide a concise summary of the events described in the video. We test on two datasets that have human-annotated captions alongside each corresponding video clip. The first dataset is YouCook2 (Zhou et al., 2018) which contains long-form cooking videos from different cuisines, where each clip segment within the video is annotated with a caption. We also test on COIN (Tang et al., 2019) which is a larger instructional dataset of household tasks and contains sentence-level text labels for each clip in the video. Given the individual video clips, we compute generation similarity metrics from the model output and the ground truth caption.

### 4.2 MODEL SETUPS

For our modeling setup, we use the modality architectures specified in the methods section 3 with the largest model variant offered and use LLaMa as our final generator. We set the few-shot examples to $n = 5$ due to the size of texts generated from the combined input modalities. In the greedy search setting, we fix the holdout and search set sizes to $|H| = |S| = 30$.

In addition to testing our few-shot approaches, we directly optimized the LLaMa weights to gauge the performance improvements given the fine-tuning computational costs. Instead of full model parameter updates, we use low-rank updates for training in this setting, as described by Hu et al. (2021). Here we are only updating the LLaMa model where the individual modality models are frozen and only provide the input modality texts used for training.

Finally, we test our approach against modality alignment strategies as done in ChatBridge (Zhao et al., 2023). ChatBridge similarly uses foundational models to encode frame and audio data. It uses a Perceiver model, which outputs the embeddings from the audio or frames into a learned fixed-sized dictionary. In this way, these embeddings can be used within LLaMa to provide additional context for the dense video embedding in conjunction with the instruction text. In the first-stage training, ChatBridge optimizes the Perceiver models by aligning each modality with ground truth text caption

| Prompt | Modalities | YouCook | | | | COIN | | | |
|---|---|---|---|---|---|---|---|---|---|
| | | BLEU-2 | BLEU-3 | METEOR | ROUGE-L | BLEU-2 | BLEU-3 | METEOR | ROUGE-L |
| Random | A | 17.71 | 9.34 | 32.37 | 35.47 | 1.33 | 0.22 | 7.09 | 11.88 |
| | AL | 18.00 | 9.15 | 33.82 | 35.61 | 1.57 | 0.44 | 9.22 | 15.84 |
| | ALI | 18.57 | 9.44 | 33.63 | 37.38 | 11.16 | 3.39 | 23.01 | 31.10 |
| | ALIO | 16.79 | 7.90 | 30.76 | 34.78 | 11.56 | 3.36 | 23.96 | 31.85 |
| Search | A | 16.66 | 8.52 | 29.90 | 34.89 | 4.22 | 0.91 | 11.85 | 16.80 |
| | AL | 18.70 | 9.30 | 33.13 | 36.77 | 9.29 | 2.68 | 22.33 | 26.75 |
| | ALI | **20.62** | **10.38** | **37.41** | **38.68** | **12.44** | **4.56** | **24.82** | **33.32** |
| | ALIO | 17.74 | 8.66 | 32.49 | 36.36 | 11.38 | 3.14 | 23.75 | 31.73 |

Table 1: PP-VLM generation performance on Youcook and COIN datasets under the few-shot setting. For few-shot inputs, we test by randomly selecting sample modalities versus using a search strategy. The modalities we test are the automatic speech recognition outputs (A), the label of the video (L), the image caption (I) as well as objects detected from the scene (O).

data. In second-stage training, it optimizes the base modality models given an instruction fine-tuning set. In the first and second training stages we align the Perceiver and the modality models to the ground truth training captions respectively.

## 5 RESULTS

We analyze how PP-VLM performs across YouCook and COIN under different modalities, model sizes, and the corresponding computation costs of our few-shot search strategies. Then we ablate the best model setups for ChatBridge with respect to different input prompts and training stages. We finally compare the results of our proposed few-shot method to fine-tuning methods.

### 5.1 PP-VLM GENERATION PERFORMANCE

**Effect of Modalities** We show the caption generation results on YouCook and COIN using our method in Table 1 using LLaMa-30B. From the results in both settings, we see that adding corresponding modalities generally helps the performance. This increase in performance occurs when we cumulatively add the ASR audio captions, the video topic label, and the image frame caption from the middle of the clip. Adding the objects obtained from the image with the other modalities doesn't improve performance. Further inspection shows that using a SAM-based method for the extraction of objects in amateur videos leads to many caption artifacts which detract from generating a relevant video caption. We also tested using object labels with SemanticSAM (Li et al., 2023a), which uses an objective for more granular segmentation descriptions but saw similar results. This indicates that to benefit from object detections, a domain-specific trained detector would be helpful, or a secondary filtering process to determine if the objects detected are relevant in context. Therefore, for the remainder of our experiments, we omit the object input modality unless indicated otherwise.

**Effect of Model Size** We further investigate how different datasets perform under varying model sizes. We evaluate under the random search setting over 5 trial runs which best represents performance when manually curating input prompts for a domain task. From the results in Figure 3 there are different patterns for each dataset we tested.

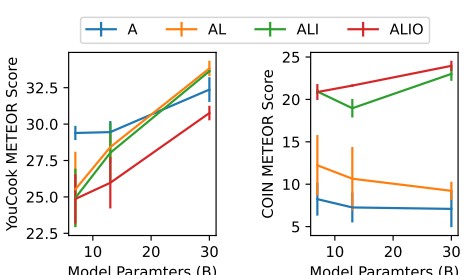

Figure 3: YouCook random sampling performance across different data modality inputs and model sizes.

In YouCook, adding modalities improves performance as the base LLM gets larger. This is counterintuitive since leveraging additional information would provide more context when working with smaller models. Since the ASR has the highest correlation with the expected caption, the smaller models use cues from this modality best. For specific domains like cook-

| Prompt | $\%\Delta\,Y\to C$ | $\%\Delta\,C\to Y$ | Cat=Furniture | | | Cat=Electical | | | Cat=Vehicle | | |
|---|---|---|---|---|---|---|---|---|---|---|---|
| | | | Cat | All-Cat | All | Cat | All-Cat | All | Cat | All-Cat | All |
| Random | -13.16 | **-38.09** | 25.32 | 24.24 | 24.73 | 30.91 | 26.08 | 26.22 | 30.89 | 26.85 | 26.75 |
| Search | **-10.43** | -45.49 | **31.37** | 26.18 | 28.18 | **32.54** | 23.60 | 29.45 | **29.78** | 25.00 | 27.03 |

Table 2: On the left we compare the performance of using prompts and trained models from YouCook (Y) to test on COIN (C) and vice versa. On the right, we ablate the validation performances in subcategories (Cat) of data in COIN. In this setup, we search/train on the same category only (Cat), all categories besides the test category (All-Cat), and all categories (All). Here we report the METEOR scores and use ALI modalities for PP-VLM.

ing, this is relevant since there is a precise set of ingredients for each step in the recipe, not commonly seen in large-scale pre-training data. Additional labels and image captions provide higher-level context but require a background understanding of the cooking activity and video, which are better encoded by texts learned by larger models.

In COIN we see that the performance is similar across model sizes. Since videos in this dataset contain common objects around the house, each modality provides the same relative value added with respect to other modalities. When considering the domain it is important to estimate *how much of the desired generated data is seen in pre-training* and to pick the combination of *modalities and model sizes* accordingly.

**Effect of Data and Compute** We are also interested in model performance as a function of domain samples available. This allows us to *estimate model performance gains for additional data samples curated* which allows for budgeting data curation and compute requirements. To analyze this, we compute the percentage improvement in METEOR performance from random selection to greedy search. We repeat this for different numbers of shots on LLaMa-13B for both COIN and YouCook, presented in Figure 4. For both datasets, we confirm that the value added for each new sample decreases as the number of shots increases. We regress values for an inverse performance curve to show how this trend continues after our tested 5-shots. This interpolation provides us with performance gain estimates for the additional training data required. For example, if

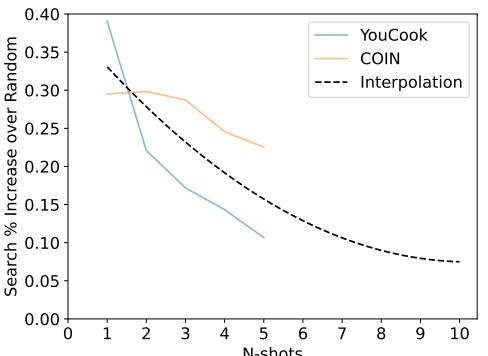

Figure 4: METEOR performance gains using greedy search over random few-shot sampling.

we wanted to perform 6-shot inference, we would need an additional new $S = |30|$ samples for the search set and would yield a 12% relative gain over if we just added a single random sample.

For search-based few-shot prompting, the compute costs are lower than full fine-tuning on training data. For LLaMa-13B each generated summary takes on average 2 seconds given transcribed audio, topic, and image inputs using a batch size of 4 on 4xA40 48GB GPUs. Therefore, for 5-shot search, it takes 2.5 hours to run an evaluation over each search sample per shot. Here the limitation is the availability of good-quality training data, which is a cost bottleneck to improving the performance metrics.

## 5.2 FEW-SHOT GENERALIZABILITY

Large foundational models are built and deployed in mind to be used in a large variety of domains. However, for applications in specific domains of interest, we analyze how transferable few-shot prompts are between domains. Furthermore, we analyze which data subsets to select for search when performance is evaluated on specific domain data.

| Training | Input Type | YouCook | | | | COIN | | | |
|---|---|---|---|---|---|---|---|---|---|
| | | BLEU-2 | BLEU-3 | METEOR | ROUGE-L | BLEU-2 | BLEU-3 | METEOR | ROUGE-L |
| Pre-trained | OS | 1.04 | 0.14 | 11.12 | 9.65 | 0.56 | 0.06 | 10.18 | 7.48 |
| | IM | 6.24 | 2.45 | 20.74 | 17.77 | 3.34 | 0.60 | 18.16 | 14.97 |
| | OS+IM | 4.31 | 1.41 | 16.95 | 14.93 | 2.45 | 0.42 | 16.37 | 12.80 |
| | FS | 2.54 | 0.81 | 12.48 | 11.11 | 2.24 | 0.43 | 12.69 | 11.13 |
| FT Stg 1 | OS | 0.97 | 0.23 | 11.40 | 10.71 | 0.11 | 0.02 | 9.31 | 7.65 |
| | IM | **9.60** | 4.20 | **27.16** | **23.55** | 1.60 | 0.24 | 13.26 | 10.16 |
| | OS+IM | 6.85 | 2.77 | 22.52 | 18.88 | 2.00 | 0.30 | 16.22 | 12.24 |
| | FS | 9.53 | **4.38** | 23.87 | 23.31 | 0.89 | 0.15 | 10.92 | 9.04 |
| FT Stg 1+2 | OS | 1.11 | 0.57 | 10.52 | 9.97 | 5.25 | 2.65 | 13.19 | 19.45 |
| | IM | 5.08 | 1.92 | 13.04 | 15.28 | **8.95** | **4.91** | **19.55** | **22.07** |
| | OS+IM | 2.37 | 0.65 | 9.11 | 9.71 | 0.96 | 0.18 | 11.09 | 9.71 |
| | FS | 6.05 | 2.11 | 18.52 | 17.62 | 1.07 | 0.24 | 10.67 | 9.10 |

Table 3: We test ChatBridge at different training stages and with different inputs. We test providing a prompt along with a few output samples (OS) of expected summaries, the relevant input modality (IM) text data for the clip, and the same few-shot (FS) inputs we use in PP-VLM. We test the pre-trained weights, stage 1 fine-tuning, and stage 1+2 fine-tuning for each dataset.

### 5.2.1 INTER-DATASET TRANSFER

To analyze the transfer capability of these models between domains, we test how PP-VLM optimized for YouCook performs on COIN and vice versa. The characteristics of both datasets are important in this task. YouCook has videos only regarding cooking but typically has dense audio captions and a larger variance in human captions. COIN is a dataset composed of diverse tasks, where audio captions are more sparse and have a smaller variance of output captions.

The transfer capability of these models between YouCook and COIN is shown in the first column of Table 2. We find that YouCook prompts transferred to COIN more gracefully than from COIN to YouCook. It is evident that *smaller, but higher quality sample annotations* provide more generalizable results between domains when searching few shot prompts.

### 5.2.2 INTRA-COIN RESULTS ANALYSIS

Beyond the transfer capability of entire datasets between domains, we further analyze the performance within subsets of COIN categories in columns 2 through 4 of Table 2. We evaluate the strategy of leveraging the higher variance in modalities present within large-scale datasets for generalizable few-shot performance. In particular, we test three subsets of categories within COIN: furniture, electrical, and vehicle videos. Within each category, we test the evaluation performance of that category under three different training settings. The first is if we only use training samples from that category. The second is if we use all training samples besides the one in that category, which replicates transfer learning performance. Third is our original setting where data from all categories are used.

We observe that the few-shot case benefits most from just using domain data when evaluating that domain. Providing searches over all categories performs closely in select instances, but since we do not fine-tune, a search limited to domains of interest provides the best results over searching through a broader dataset. Searching over a large set where the category is not present leads to the latest drop in performance for that category, and reflects the domain transfer performance between YouCook and COIN.

### 5.3 CHATBRIDGE ALIGNMENT

After investigating the properties of PP-VLM, we compare how training-based methods like Chat-Bridge perform on these summarization tasks. ChatBridge has two stages of training: stage 1 modality alignment and stage 2 fine-tuning. We perform stage 1 fine-tuning where we align the frozen image and audio models to generate captions for the corresponding clip by training the Perceiver models. We sample 16 frames from each clip to pool for the video token representation vector. For stage 2 fine-tuning, we tune over the same output captions but optimize the base image and audio models in this stage while the Perceiver is frozen.

In addition to the different stages of training, we also test different input prompts to ChatBridge. Since it uses Vicuna-13B (Chiang et al., 2023) for generation over conversational tasks, we explore how to best query the model for our summarization tasks. We prompt the model to summarize the clip using the audio and video embeddings, and also provide 5 output samples (OS) for reference so it can match the style of short summaries. Instead of just asking the model to summarize the clip given only the video and audio embeddings, we also test how useful adding text modality information is. This is done by providing text input modalities (IM) for the input clip as the prompt. Another setting we test combines both the reference output samples and input modalities (OS+IM). Finally, we test the same few-shot (FS) inputs that we use in PP-VLM. The results for these input settings across training stages are presented in Table 3.

In the results, we can see the initial performance on the pre-trained checkpoint, which is the most comparable setting to PP-VLM as no fine-tuning has been performed. Like in our PP-VLM analysis, we see different tuning strategies given each dataset. In stage 1, the larger improvements in performance were made in YouCook while COIN provided similar results. Since YouCook specializes in specific cooking ingredients, updating a minimally parameterized Perceiver improved the performance, while a more diverse dataset such as COIN couldn't leverage such gains. However, in Stage 2 the larger modality models tend to overfit the smaller YouCook dataset (1.3k training videos), leading to a drop in hold-out performance. In contrast, learning better audio and video frame representation in COIN provided a boost in performance (9k training videos). Across training stages, we observe that adding the input modalities(IM) to the prompt provided the best performance, validating the utility of modalities as text inputs.

### 5.4 PLUG-AND-PLAY FEW-SHOT VERSUS MODEL TUNING

Given the exploration of ChatBridge setups, we compare the performance against our PP-VLM few-shot search schemes. For PP-VLM we compare our random search strategy, which reflects the setting where new domain samples are provided to the model, rather than having a large dataset to search for samples over. We also perform LoRA fine-tuning on LLaMa where we use video text input modalities (IM), and the caption as the expected output. This allows us to directly compare fine-tuning versus few-shot performance for our summarization task given only text inputs.

From the results in Table 4, we can see that adding additional shot samples helps PP-VLM performance. LoRA fine-tuning is comparable to our 2-3 shot methods. This indicates the strength of in-context learning within foundational models for summarization tasks, where only a few demonstrations are necessary. For ChatBridge we compare the best input modality (IM) setting we determined from our previous evaluation at different stages of training, which performs comparable to our random 4-5 shot methods. When using greedy search (S:5), which uses less training data and compute, our

| Model | setup | YouCook | COIN |
|---|---|---|---|
| PP-VLM 13B | R:0 | 16.88 | 17.66 |
| | R:1 | 19.48 | 17.63 |
| | R:2 | 23.76 | 17.73 |
| | R:3 | 26.64 | 18.87 |
| | R:4 | 28.06 | **21.51** |
| | R:5 | **29.79** | 20.06 |
| | S:5 | 32.97 | 24.58 |
| | LoRA | 25.69 | 17.18 |
| ChatBridge 13B | PT | 20.74 | 18.16 |
| | S1 | 27.16 | 13.26 |
| | S1+2 | 13.04 | 19.55 |
| PP-VLM 30B | R:5 | 33.63 | 23.01 |
| | S:5 | 37.41 | 24.82 |

Table 4: We compare METEOR performance of PP-VLM under few-shot conditions using ALI under random sampling n samples (R:n) versus search sampling (S:n), LoRA and fine-tuned ChatBridge. ChatBridge setups use the text input modality where results are reported across the pre-trained model (PT), stage 1 (S1), and stage 1+2 (S1+2) fine-tuning.

PP-VLM is able to provide better quality summaries. Notably for pre-trained ChatBridge, it outperforms 0-shot PP-VLM, which indicates that the combination of *latent modality embeddings* and *semantic text embeddings* can be useful in hybrid methods. These can be tasks beyond summarization where the intermediate modality outputs cannot be explicitly expressed as text from a single frame, such as learning cause and effect, or temporal dynamics of a subject.

## 6 CONCLUSION AND FUTURE WORK

Foundational models from multiple modalities leverage large-scale data to achieve SOTA performance within their domains. Due to this, leveraging foundational models from multiple modalities has been an active area of research for inputs that leverage this multi-modality, such as videos. These video models use the corresponding video and audio inputs to carry out instructions prompted by the user. To leverage the embeddings of these modalities, each input modality is aligned to share the same embedding space. Then the individual modality models and the language model can be further fine-tuned for a specific task. Instead of leveraging latent modality embeddings, we test using the text descriptions of these modalities directly. This plug-and-play approach allows using off-the-shelf generative models and composes their outputs within an LLM to carry out an instruction. Testing few-shot approaches to adapt this model to video summarization tasks reduces computation cost and data overhead needed by fine-tuning models. Our results provide key insights in:

- Understanding the domain of the videos when selecting which modalities and model sizes to use for few-shot modeling.
- Evaluation of the value added by obtaining more data and compute requirements when testing different few-shot evaluation strategies.
- Transferring few-shot prompts between domains works best with high-variance captions, and performance improvement within a specific domain only requires few-shot samples from that domain.
- Showing what stages of modality fine-tuning are best given the size of the training corpus and which text modality prompts to use.

In future work, we want to leverage PP-VLM to incorporate more information beyond just the input video text modalities. External documents or knowledge graph information can provide greater context when performing video tasks. For example, recipe data or instructional steps can help improve the performance of YouCook and COIN captioning respectively. In longer-form videos composed of multiple sub-clips, we want to investigate how to pass information between clips to provide a better context of the video as a whole. This involves deciding which information is most useful to share between video clips. Proving such external knowledge helps the video model bridge more complicated tasks beyond input summarization, such as predicting future video events or conversational tasks.

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
