# OpenReview forum: "Exploring Efficient Foundational Multi-modal Models for Video Summarization"
_ICLR.cc/2024/Conference — ICLR 2024 Conference Withdrawn Submission_

### Official Review · Reviewer_WPWU · 2023-10-25

**Soundness:** 2 fair
**Presentation:** 3 good
**Contribution:** 2 fair
**Rating:** 3
**Confidence:** 4

**Summary:**

This work study the generalizability and efficiency of leveraging foundation multi-modal models for video summarization task. Specifically, a plug-and-play style video language model is proposed, where an LLM uses the text generated from each multi-modal models jointly as prompt to generated a textual output. The design of the framework allow the users to avoid model alignment on feature space.

**Strengths:**

+ Explore a new way to leverage foundation multi-modal model without the needs to align the corresponding feature to the same embedding space.
+ few-shot instruction adaptation strategies to fine-tune an LLM
+ The evaluation on computational cost, insights discussion, and validation on domain shift scenario present practical insights under specific hardware and data constraints.

**Weaknesses:**

- The Ablation of the effect of modalities is incomplete. The modalities are added in a cumulative manner. The design place a strong emphasis on the ASR audio caption, which provide strong information to generate a summary. This is particular "problematic" when both dataset are instructional dataset. Hence, it is good to provide results with more variants (e.g., simply remove one modality each time from full model).
- It is necessary to evaluate the proposed approach on non-instructional dataset.

**Questions:**

- For Table 2, can the author provide the METEOR scores for Y -> C? This can give a more intuitive comparison between inter-dataset and intra-dataset transfer.

---

### Official Review · Reviewer_PkUQ · 2023-10-28

**Soundness:** 2 fair
**Presentation:** 2 fair
**Contribution:** 1 poor
**Rating:** 3
**Confidence:** 4

**Summary:**

This paper proposed a plug-and-play video language model by using the texts generated from each input modality in the language mode.

**Strengths:**

(1) The paper formulation is good and clear.

(2) The method is straightforward and easy to understand.

**Weaknesses:**

(1) The foundation of this work lies in image-to-text generation via BLIP. The resulting text, in conjunction with certain transcripts and metadata, is fed into LLaMa to produce the textual output. The outcome heavily relies on the performance of BLIP.

(2) Exploring ablation studies that focus on specific components could elucidate their importance. Additionally, what if we exclusively use transcripts to generate summaries? This avenue warrants investigation.

(3) While the system generates textual summaries, what about video summaries, such as keyframes? This aspect remains unaddressed.

(4) The results presented in Table 1 suggest that incorporating (O) does not provide any discernible benefit.

(5) To enhance the study's depth, it would be valuable to include more baseline models for comprehensive comparison.

**Questions:**

Please see the comments above.

---

### Official Review · Reviewer_F78M · 2023-10-29

**Soundness:** 2 fair
**Presentation:** 2 fair
**Contribution:** 1 poor
**Rating:** 3
**Confidence:** 3

**Summary:**

This paper proposes a foundation model for video summarization. It does not do alignments between different metadata in the videos, such as audio, images, and captions. Instead, it leverages other existing models, such as SAM, Word2Vec, BLIP2 to convert the information into text. Then, these text modalities are fed into the LLM, i.e., LLaMa to generate the summarization of the videos. In this framework, the authors claim that it achieves good generalization ability without introducing much computational cost.

**Strengths:**

[1] Originality: Since videos themselves contain different modalities, it is pretty important to develop a model that can unify different metadata with different modalities to do different video tasks.

[2] Quality: This paper evaluates their method on different datasets, such as YouCook and COIN.

**Weaknesses:**

[1] Originality: The most essential concern is originality. This paper proposes linking different methods to convert multimodal signals to text signals. Therefore, these components are not novelties. The author claimed the novelty is the whole framework. However, this framework is very similar to different Multimodal Large Language Models, such as LLAVA, PandaGPT, but even in more complex design.

[2] Originality: The second and the third claimed novelty are experimental results or ablation studies. It can hardly be claimed as the contribution of the paper.

[3] The experimental results are not comprehensive enough. Since the developed method is largely designed for video summarization, it can not be claimed as a foundation model. A foundation model should benefit diverse video tasks, such as action recognition, temporal action proposal, spatiotemporal detection, etc.

**Questions:**

Please see the weaknesses.

---

### Official Review · Reviewer_zhmP · 2023-10-31

**Soundness:** 2 fair
**Presentation:** 2 fair
**Contribution:** 1 poor
**Rating:** 3
**Confidence:** 4

**Summary:**

The paper presents a framework for video summarization using language models. This framework extracts information from the video using off-the-shelf, modality-specific models and converts it into text. This textual information is then used to populate a predefined template prompt, which the language model utilizes to summarize the video. The authors assess their model in both zero-shot and few-shot settings across two datasets. Additionally, the paper explores the impact of each modality on summarization quality, finding that audio transcripts are the most significant for the evaluated datasets. The authors also investigate how well the prompts tailored for one dataset can be applied to another.

**Strengths:**

1. The paper studies an important problem which is how to perform video summarization using Large Language Models.
2. The authors propose evaluating the method under both zero-shot and few-shot settings, a reasonable approach considering the prevailing trend of generalist models tackling a broad spectrum of tasks.
3. The paper examines a greedy selection method to determine the best samples in a few-shot setting.

**Weaknesses:**

1. The paper's main weakness is overclaiming contributions that are not well supported.
    - The second contribution highlights:
"what input modalities and model sizes are fitting for a given domain, and expectations of how multi-modal models adapted to one domain will fare in another." This claim seems overly broad, especially since proving such a statement would presumably require experiments across multiple tasks, datasets, and domains. However, the paper restricts its experiments to video summarization using just two datasets.
     - The third contribution suggests a generic method to fine-tune multimodal LLMs. In reality, the paper only offers a tailored approach for video summarization. How can this method be applied to non-local tasks like Action Localization, Video Object Segmentation, Video Action Segmentation, Dense Captioning, and so forth? The paper's claims far exceed its actual content, which essentially presents a method for using LLMs specifically for video summarization.

2. The method's focus on video summarization is ambiguous. The paper emphasizes that only the center frame is considered. So, is this method merely an advanced image captioning technique? Without the audio modality, the approach essentially becomes an imaging technique. To address this assumption, one would expect the method to perform competently even without audio (as inferred from Table 1).
    - Related to this point the selection of the dataset is key. Why didn't the authors use other summarization datasets like TvSum? Or even better, other tasks like VQA that require more temporal reasoning.
    - Since this paper does not reason over temporal sequences, isn't it more like an enhanced image captioner?
    - There are similar works trying to use LLMs to get video descriptions [G], what is the advantage of the current work over them?

3. Last but not least, the proposed method doesn't introduce a novel approach. Several papers have already explored converting modalities into text and utilizing LLMs for video tasks. For example, Yang et. al. (AAAI-2022) [A] and Tiong, et. al. (ACL-2022) [B] presented a similar framework, wherein frozen LLMs processed the outputs of an image captioner for VQA. With numerous analogous methods [C,D,E,F] introduced recently, the originality of this work is not evident to me.


[A] Yang, Z., Gan, Z., Wang, J., Hu, X., Lu, Y., Liu, Z., & Wang, L. (2022, June). An empirical study of gpt-3 for few-shot knowledge-based vqa. In Proceedings of the AAAI Conference on Artificial Intelligence (Vol. 36, No. 3, pp. 3081-3089).

[B] Tiong, A. M. H., Li, J., Li, B., Savarese, S., & Hoi, S. C. (2022). Plug-and-play vqa: Zero-shot vqa by conjoining large pretrained models with zero training. arXiv preprint arXiv:2210.08773.

[C] Guo, J., Li, J., Li, D., Tiong, A. M. H., Li, B., Tao, D., & Hoi, S. (2023). From Images to Textual Prompts: Zero-shot Visual Question Answering with Frozen Large Language Models. In Proceedings of the IEEE/CVF Conference on Computer Vision and Pattern Recognition (pp. 10867-10877).

[D] Li, C., Ge, Y., Mao, J., Li, D., & Shan, Y. (2023). TagGPT: Large Language Models are Zero-shot Multimodal Taggers. arXiv preprint arXiv:2304.03022.

[E] Pan, J., Lin, Z., Ge, Y., Zhu, X., Zhang, R., Wang, Y., ... & Li, H. (2023). Retrieving-to-Answer: Zero-Shot Video Question Answering with Frozen Large Language Models. arXiv preprint arXiv:2306.11732.

[F] Chen, J., Zhu, D., Haydarov, K., Li, X., & Elhoseiny, M. (2023). Video chatcaptioner: Towards the enriched spatiotemporal descriptions. arXiv preprint arXiv:2304.04227.

**Questions:**

- What are the genuine contributions of this paper? Instead of overstating, I'd recommend presenting only those contributions substantiated by the experiments. It's acceptable if the contributions are task-specific and not generalized.
- How does this method qualify as a video summarization technique rather than merely an enhanced captioner? (refer to the second weakness)
    - What would the outcome be if Audio isn't a provided modality, or if the audio lacks speech content? How effective would this method be in summarizing silent videos?
- What novel insights or methods does this paper introduce? (as mentioned in the third weakness)